

# Performance in a novel environment subject to ghost competition

Karen Bisschop[1,2], Frederik Mortier[2], Dries Bonte[2] and
Rampal S. Etienne[1]

[1] Groningen Institute for Evolutionary Life Sciences, University of Groningen, Groningen,
The Netherlands
[2] Department of Biology, Universiteit Gent, Ghent, Belgium

## ABSTRACT

**Background:** A central tenet of the evolutionary theory of communities is that competition impacts evolutionary processes such as local adaptation. Species in a community exert a selection pressure on other species and may drive them to extinction. We know, however, very little about the influence of unsuccessful or ghost species on the evolutionary dynamics within the community.
**Methods:** Here we report the long-term influence of a ghost competitor on the performance of a more successful species using experimental evolution. We transferred the spider mite *Tetranychus urticae* onto a novel host plant under initial presence or absence of a competing species, the congeneric mite *T. ludeni*.
**Results:** The competitor species, *T. ludeni*, unintentionally went extinct soon after the start of the experiment, but we nevertheless completed the experiment and found that the early competitive pressure of this ghost competitor positively affected the performance (i.e., fecundity) of the surviving species, *T. urticae*. This effect on *T. urticae* lasted for at least 25 generations.
**Discussion:** Our study suggests that early experienced selection pressures can exert a persistent evolutionary signal on species' performance in novel environments.

## INTRODUCTION

Populations are facing a continuously changing world that they can possibly cope with in various ways, such as through phenotypic plasticity or by tracking their favoured habitat. If these solutions are not possible, evolutionary rescue by genetic adaptation may eventually allow persistence (*Lindsey et al., 2013*). One factor influencing this local adaptation is competition.

Interspecific competition is known to influence local adaptation in many different ways, but the effect is still largely unpredictable (*Rice & Knapp, 2008*; *Alzate et al., 2017*; *Zhao et al., 2018*). First, heterospecific competitors might modify the selection pressure exerted by the abiotic environment, enhancing or limiting genetic adaptation to the novel environment (*Osmond & De Mazancourt, 2013*). Classical examples of enhanced genetic

Corresponding author
Karen Bisschop,
kbisschop.evo@gmail.com

adaptation are seen in adaptive radiations of three-spined sticklebacks or fast character displacements in Darwin finches or Myzomelid honeyeaters (*Diamond et al., 1989*; *Schluter, 1994*; *Reznick & Ghalambor, 2001*). Previously, we found that additional selection pressure exerted by a congeneric species (*T. evansi*) facilitated adaptation of the focal species (*T. urticae*) to a novel environment under high dispersal from a maladapted ancestral population (*Alzate et al., 2017*). Adaptation to the novel environment can also be reduced by interspecific competition when there is, for instance, a trade-off between traits responsible for adaptation to the competing species and to the novel environment (*Chesson, 2000*; *Siepielski et al., 2016*).

Furthermore, interspecific competition can create new niches or change the current environment for species to adapt to. Species may use waste products or adapt to plants with modified defences caused by co-occurring individuals (*Sarmento et al., 2011*; *Lawrence et al., 2012*). These new niches will subsequently create opportunities for adaptive shifts to novel environmental conditions. This illustrates that competition and facilitation can jointly shape evolution, making it difficult to study the consequences of interspecific competition alone.

As a last scenario, interspecific competition can hinder or limit the process of local adaptation by restricting resource availability and hence decrease effective population size. The resulting increased probabilities of genetic drift will then decrease the evolutionary potential and hence the chance of local adaptation (*Lawrence et al., 2012*; *Osmond & De Mazancourt, 2013*; *Zhao et al., 2018*). Extreme hindrance to local adaptation can even cause extinction of one of the species (*Jaeger, 1970*; *Bengtsson, 1989*; *Griffis & Jaeger, 1998*).

In this study we aimed to further unravel the effect of interspecific competition on adaptation to a novel food source. We therefore performed an evolutionary experiment with two related spider mite species, *Tetranychus urticae* and *T. ludeni*, adapting to a novel host. Both species were placed alone or together on a new host plant and we wanted to study how this interspecific competition affects local adaptation. More precisely, we wanted to test for differences in the rates of adaptation and the final performance on the novel food source at the end of the experiment. The two competitors were supposed to be competitively similar, but the experiment demonstrated that this was not the case: both species could only temporarily co-occur, because *T. ludeni* went extinct after a few generations. We nevertheless continued the experiment and conducted subsequent analyses that provided convincing evidence for the effect of the ghost competition on adaptation. More precisely, we focused on the surviving species, *T. urticae*, and explored whether we could detect long-term evolutionary effects on performance (measured as fecundity) of this species due to differences in early selection pressures caused by the ghost competitor, *T. ludeni*. We chose the average of the initial population size of the unsuccessful species during the 1st month of co-occurrence as an indication for the early competitive pressure. These differences in population sizes arose naturally and can be attributed to selection, as well as drift and founder effects.

While inferior competitors are predicted to eventually go extinct, they may co-occur with the more successful competitors for many generations (*Holmes & Wilson, 1998*; *Lankau, 2011*). These early and non-persisting interactions may leave a strong signature

on the future community dynamics (*Law & Daniel Morton, 1996*; *Miller, TerHorst & Burns, 2009*; *Mallon et al., 2018*), because they have the possibility to induce large habitat modifications or evolutionary changes in the more successful species. Historical contingency (i.e. the influence of the arrival time of a certain species in a community; *Fukami, 2015*) in terms of limitations imposed by so-called ghost species (*Hawkes & Keitt, 2015*) may thus have a strong impact on the eco-evolutionary trajectories of populations and communities, in the same way as successful species do (*Fukami, 2015*). The role of competition intensity of an inferior species prior to its extinction on the ecological and evolutionary dynamics of persisting species is still largely unknown, however.

In this study we found a lower fecundity of *T. ludeni* on bean and cucumber than *T. urticae* in the control populations, which may explain their rapid extinction. Still, the ghost species *T. ludeni* showed an effect on the surviving species *T. urticae*, because the eventually achieved strength of adaptation of *T. urticae* increased with the initial density of *T. ludeni*. We therefore suggest that ghost competition may lead to differences in long-term local adaptation.

## MATERIALS AND METHODS

### Study species

We used two species of the family Tetranychidae (Acari, Arachnida): *Tetranychus urticae* Koch, 1836, and *T. ludeni* Zacher, 1913. These herbivorous mite species are highly suitable for evolutionary experiments due to their small body sizes, their possibility to maintain large populations in the lab, and short generation times (*Zhang, 2003*).

For this study, we used different inbred populations of *T. urticae* from *Bitume et al. (2013)*. Each population originated from two adult females from the LS-VL line (*Van Leeuwen, Stillatus & Tirry, 2004*) and was afterwards kept at low population densities. The LS-VL line was collected from roses in October 2000 (Ghent, Belgium, Europe). After this initial collection, all populations were maintained on bean plants (*Phaseolus vulgaris*, Prelude).

We used two populations of *T. ludeni*: the Tl Alval (Lisbon, Portugal) and Tl CVM (Lourinhã, Portugal). Both populations were sampled early autumn 2013 from common morning-glory and afterwards maintained on bean plants (*P. vulgaris*, Prelude). The founder populations were 160 and 300 individuals for Tl Alval and Tl CVM respectively. Our evolutionary experiment started in September 2015, implying that *T. urticae* and *T. ludeni* had been under laboratory conditions for approximately 15 and 2 years, respectively.

For this study, we chose to subject different inbred lines of *T. urticae* that were created in 2012 (*Bitume et al., 2013*) to further inbreeding by mother-son mating for one more generation prior to the experiments. This resulted in the creation of 13 isofemale lines. The genetic variation within the lines is therefore very low, but larger among lines as we used different inbred lines from *Bitume et al. (2013)*. It may sound counterintuitive to use inbred populations for an evolutionary experiment that mainly uses standing genetic variation (note that no spider mites were added during our experiment), but in this way we could generate genetically similar replicates and control for putative initial drift effects by

differences in starting genetic variation. We deemed this more important than potential inbreeding effects, because no effects of inbreeding on genetic trait variation have been found in these and other lines (*Van Petegem et al., 2018*). Furthermore, we created six isofemale lines for *T. ludeni* (coming from Tl Alval and Tl CVM). We wanted to create 13 lines for this species as well, but were unsuccessful due to low fertility or early mortality. In hindsight, this may partly explain the rapid extinction *T. ludeni*: even the six initially surviving lines were probably far from optimal.

We created a control population of *T. ludeni* from the stock (Tl Alval and Tl CVM) and placed them on bean plants (four two-weeks-old plants). We created a control population of *T. urticae* from the created 13 isofemale lines (four mites per line) and placed them also on bean plants (four 2-weeks-old plants). All populations were kept in a climate-controlled room (25–30 °C, 16:8 L:D).

## Experimental set-up

Novel host islands were created by placing two 3-weeks-old cucumber plants, *Cucumis sativus* Tanja, in boxes with yellow sticky paper (Pherobank) at the bottom and Vaseline at the walls to avoid contamination between islands; this method is known to work from previous research (*Alzate et al., 2017*; *Alzate, Etienne & Bonte, 2019*; *Bisschop et al., 2019*). These units consisting of multiple plants are referred to as 'islands' because they represent isolated habitat. They represent continuous habitat because the leaves were overlapping allowing easy dispersal across the plants. After the 1st week, two fresh 3-weeks-old cucumber plants were added to create the island size of the experiment. To provide enough food for the spider mites, the islands were then weekly refreshed by replacing the two oldest plants with two new 3-weeks-old cucumber plants. In this way, sufficient time was provided for a generation of spider mites to develop on the new plants, while allowing the population to move toward the fresh leaves. The removed old plants may have contained mites or unhatched eggs, but we nevertheless chose this refreshment procedure to maintain relatively natural movement dynamics. It is for instance known that especially young fertilised females disperse more (*Li & Margolies, 1993*) and dispersive individuals may differ in their body condition or performance compared to sedentary individuals (*Bonte et al., 2014*; *Dahirel et al., 2019*). This refreshment procedure may have caused an extra competitive pressure if one species was more dispersive or delayed its dispersal for avoiding competition, but we preferred to design the experiment in a way that it resembled more the actual life strategy of spider mites (colonisation with few founders followed by rapid growth).

The spider mite isofemale lines (as subscribed under 'study species') were placed on the islands with or without heterospecifics. Eight replicate islands received both *T. urticae* and *T. ludeni*. Eight replicate islands received only *T. urticae* and another eight replicate islands received only *T. ludeni*. Each island started with the same total population size of 52 adult females and as similar as possible gene pool. Therefore, the group with only *T. urticae* received four adult females from each of the 13 isofemale lines. The group with only *T. ludeni* received four adult females from the six isofemale lines and was supplemented with 28 females from its stock population, because of the lack of success of
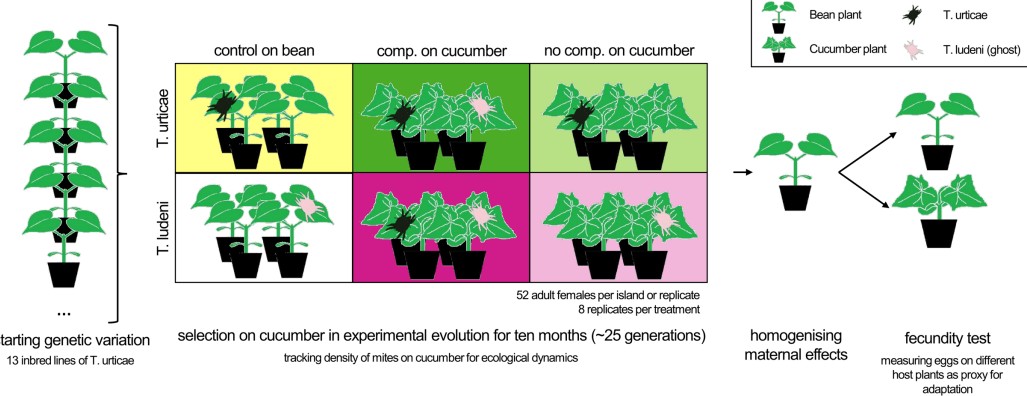

**Figure 1 The experimental set-up.** Adult females from 13 inbred lines of *T. urticae* (*Tu*) were equally divided over the different treatments to create the same starting genetic variation. Populations with *T. ludeni* (*Tl*) had a higher genetic variation as only six inbred lines were used and supplemented with the stock population. The treatments were a control population on bean plants (yellow box for *Tu* and white for *Tl*), a competition treatment with both species present on cucumber (dark green or dark pink box for *Tu* and *Tl* respectively) or a no competition treatment on cucumber (light green box for *Tu* and light pink box for *Tl*). The density of the populations of mites on cucumber was tracked for ecological dynamics and individual fecundity tests were performed on the novel and initial host plants after two generations on the initial host plant for homogenising maternal effects. The boxes have the same colours as used in Figs. 2 and 3.

creating more isofemale lines. The group with both spider mite species received 26 adult females of *T. urticae* (two from each of the isofemale lines) and 26 adult females of *T. ludeni* (11 *T. ludeni* females per island came from the six isofemale lines and were supplemented with 14 mites from its stock population). Control populations on bean were created as explained under 'study species'.

The use of the outbred stock population of *T. ludeni* to supplement the populations provided an unanticipated opportunity as it increased the initial genetic variation of *T. ludeni* among replicates and hence differences in early selection pressures on *T. urticae*.

We started with a relatively low population size to make it biologically relevant as natural populations usually colonise plants at small population sizes. All adult female mites were equally distributed over the plants.

We chose the same total population size and no differences among island sizes, as it is known that differences in densities change both the intra-and interspecific competitive pressure and that an increase in island size would change the adaptive potential of the treatment (*Alzate, Etienne & Bonte, 2019*). This necessarily meant that the initial population size for each species was not the same across the different treatments; we recognise that this may affect genetic drift and cause sampling effects, which we will come back to in the "Discussion". A schematic overview of the different treatments is provided in Fig. 1.

The total experiment lasted for 10 months, which is approximately 25 generations and long enough to detect local adaptation (*Gould, 1979*; *Fry, 1989*; *Magalhães et al., 2007*, *2009*; *Bonte et al., 2010*). For logistical reasons the experiment was performed in two blocks with 1 month difference, each block consisted of four replicate islands per treatment.

## Measurements

### Ecological dynamics (population density assessment)

Every 2 weeks, the density of the spider mites in the evolutionary experiment was measured by counting adult females on a square of $1 \times 1$ cm$^2$; the first counting was done after 2 weeks. The location of the square was chosen right next to the stalk of the highest, fully grown leaf of the two newest plants of each island, keeping in mind that the mites had one week to move from the old plants towards the fresh plants before counting. Both the abaxial as well as the adaxial side were measured and summed for a total overview. The location on the leaf was chosen to standardise the measurements in time and make them comparable. The populations of *T. ludeni* under competition with *T. urticae* went extinct after approximately 2 months. To get an impression of its competitive pressure on the more successful *T. urticae* populations while it was still present, we used the mean population density of the 1st month of *T. ludeni* (hereafter called 'early competitive pressure of the ghost competitor').

### Evolutionary dynamics (fecundity assessment)

Fecundity tests for the control populations on bean and for the experimental cucumber populations were performed every 2 months to determine the level of adaptation. As the experimental populations of *T. ludeni* went extinct under competition, we obviously only have results from fecundity tests on the control population of *T. ludeni*. We chose fecundity as a proxy of adaptation because previous research confirmed it to be the best predictor of adaptation compared to survival or development (*Magalhães et al., 2007*; *Alzate et al., 2017*; *Alzate, Etienne & Bonte, 2019*). Five adult females were sampled from each island and separately placed on a bean leaf disc ($17 \times 27$ mm$^2$) for two generations of common-garden to standardise juvenile and maternal effects (*Magalhães et al., 2011*; *Kawecki et al., 2012*). Bean discs were chosen because bean plants are very suitable host plants with a low selection pressure and will not cause a change in allele frequencies of the evolved lines (*Magalhães et al., 2011*). These leaf discs were placed in a petri dish on wet cotton wool and surrounded with paper strip borders. Then, the fecundity of two quiescent deutonymph females that originated from the same common-garden replicate was tested. One female was put on a bean leaf and one on a cucumber leaf (same set-up as for common garden) in a climate cabinet of 30 °C under 16:8 L:D. Bean and cucumber plants were grown specifically for this test (for 2 and 3 weeks respectively) and were protected against herbivory before the test. Fecundity (number of eggs laid after 6 days) was measured based on daily pictures taken. Females that drowned in the cotton before the 6th day were excluded from the analysis (this was 13.5% for the populations of *T. urticae* without *T. ludeni*, 15% for the populations of *T. urticae* with the ghost competitor, and 10.5% for the populations of *T. urticae* in the control treatment maintained on bean). The cucumber plants that were necessary for the leaf discs for the fecundity test after 4 months did not grow for one of our two experimental blocks. Therefore, we were not able to test fecundity at that time point for these replicates. In total, the fecundity was measured for 974 females (exact sample sizes per treatments are given in the electronic Table S1).

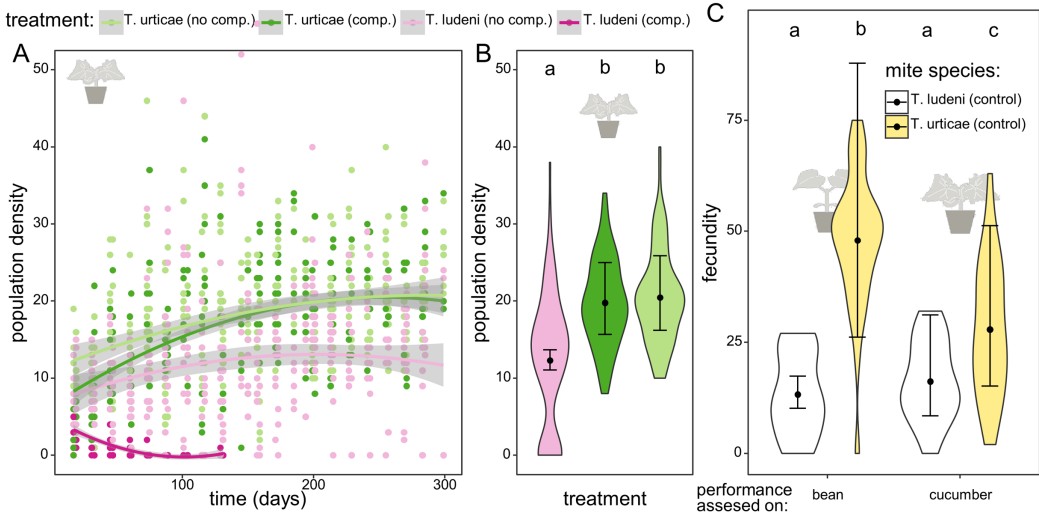

**Figure 2** **The dynamics and performance of the ghost competitor.** (A) Overview of the population density for the different treatments of the experimental populations on cucumber. Population density of *Tetranychus urticae* (green dots) and *T. ludeni* (pink dots) measured as the sum of the abaxial and adaxial density (number of adult females/cm$^2$) per island through time. The lighter colours correspond to the populations in absence of the competing species and the darker to the treatment where both species are present. The lines are smoothing curves with their respective 95% confidence interval. (B) Comparison of the densities for the different treatments at the plateau phase (starting from 200 days). The letters above the violin plots indicate the significant differences. No significant difference was found between population densities of *T. urticae* with or without initial competition, and both populations reached significantly higher densities than *T. ludeni*. The violin plots show the observed data, and the points and lines show the mean model estimates and their 95% confidence interval, respectively. (C) Comparison of the individual performance of the control populations of *T. ludeni* and *T. urticae* (both maintained on bean plants) on bean and cucumber leaf discs at the first measured time point. The fecundity of *T. ludeni* is significantly lower than that of *T. urticae*, on both bean and cucumber.

## Statistical analysis

We used general linear mixed models (GLMMs; except for the dynamics and performance of the ghost competitor where a GLM was used) with Negative Binomial distribution with log link to account for overdispersion for both the fecundity and population density measures. The variance was determined as $\mu \times (1 + \mu/k)$ in which $\mu$ is the mean and $k$ is the overdispersion parameter (standard negative binomial parametrisation) (*Hardin & Hilbe, 2007*). The violin plots in the figures illustrate the amount of overdispersion in the data.

### The dynamics and performance of the ghost competitor

We first studied the performance of the control populations that had been maintained on bean of both species on bean and cucumber. The dependent variable in the maximal model was fecundity (number of eggs after 6 days) and the explanatory categorical variables were the plant species during the fecundity assessment (bean or cucumber) and the mite species (*T. urticae* or *T. ludeni*). Model selection was based on the lowest AICc and a Wald $\chi^2$ test was performed on the maximal model to check the reliability of the model selection. We present below the results of the best-fitting model. Pairwise comparisons for the

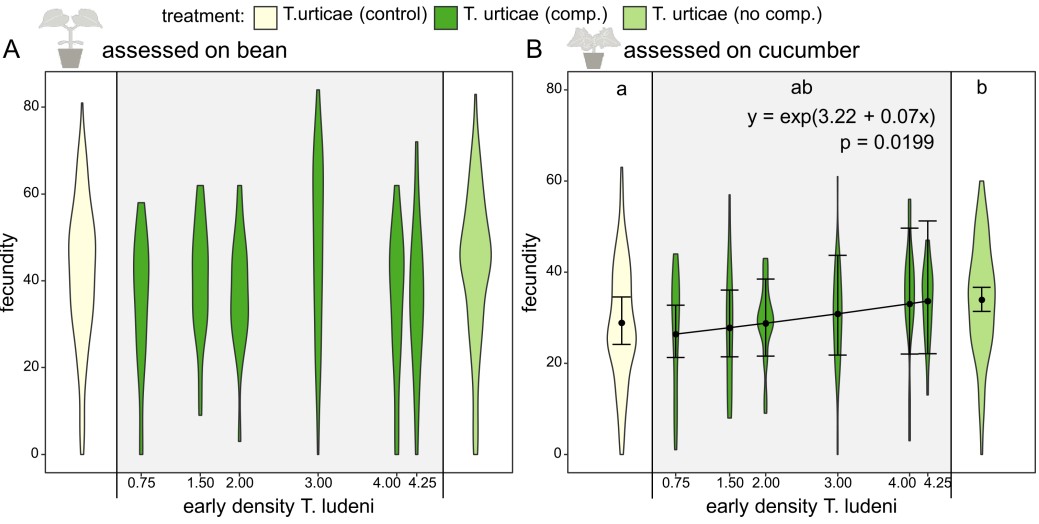

**Figure 3 Fecundity affected by ghost competition.** On the *x*-axis the different treatments (the control population of *T. urticae* from bean (yellow), *T. urticae* with ghost competition of *T. ludeni* (dark green) and *T. urticae* from cucumber but without *T. ludeni* (light green)) are presented. The scale on the *x*-axis indicates the early competitive pressure of *T. ludeni* (average number of adult females/cm² during the 1st month) for the replicates of the treatment under ghost competition; this treatment is shown in the grey box. On the *y*-axis the fecundity (number of eggs after 6 days) of *T. urticae* is presented. The variable time was not present in the best-fitting model, so we presented all data points per treatment independent of the time it was measured. (A) shows the fecundity assessed on bean, while (B) gives the results assessed on cucumber. Populations without *T. ludeni* maintained on cucumber (light green) performed significantly better on cucumber than the control (yellow) and seemed therefore locally adapted. The treatment under competition (dark green) was intermediate between both other treatments. A significant relationship between the density of *T. ludeni* (in the treatment under ghost competition; dark green) and the fecundity of *T. urticae* was found when assessed on cucumber. This indicates that early experienced selection pressures can exert a persistent evolutionary signal on species' performance in novel environments. Each violin plot presents the observed data, while the points and lines show the means of the model estimate and their 95% confidence interval, respectively.    

variables in this best-fitting model were adjusted for multiple comparisons with Tukey's method.

### *Signature of the ghost competitor on performance of T. urticae*
*Evolutionary dynamics (fecundity assessment)*

We investigated the impact of the density of *T. ludeni* and of *T. urticae* at the onset of the experiment (i.e. mean density during the 1st month) on the fecundity of *T. urticae* on its initial and novel host plant. The explanatory variables in the maximal model were time (as categorical variable; 2, 4, 6, 8 or 10 months), the density of *T. ludeni* (continuous variable), the density of *T. urticae* (continuous variable), and the interaction between time and densities of both species. In this way, we aimed to determine whether it was the own density or the density of the ghost competitor that affected performance of *T. urticae*. We compared this with an additional model with the total density (summing the density of *T. ludeni* and *T. urticae*), time, and their interaction to find out whether fecundity was affected by the species' individual densities or just the total density. The random effect for

all models was the replicate island nested within the experimental blocks. However, for the assessment of fecundity on cucumber the nestedness in the random effects led to overfitting of the full model, as the random effect variance was estimated to be zero (*Magnusson et al., 2018*). As a consequence, we only used replicate island as random variable for the assessment on cucumber. We chose a categorical variable for time instead of a continuous one, because differences in quality of leaves at the different measurements were likely and we did not want to assume a linear response of adaptation. For completeness, we also performed the analyses with time as a continuous variable; the results are similar, see (Tables S2–S6). Model selection was based on the lowest AICc and an additional Wald $\chi^2$ test was performed on the maximal model. We present the results of this best-fitting model in the main text. Pairwise comparisons for the slopes and means in this best-fitting model were adjusted for multiple comparisons with Tukey's method (summary statistics for full models are provided in the electronic Table S7).

### Ecological dynamics (population density assessment)

We used the density assessed through time of the different spider mite populations to investigate differences in demography between the species with or without competitor after the plateau phase (starting from day 200 based on visual inspection). The dependent variable in the maximal model was the density (number of adult female mites per cm$^2$) and the explanatory variable was the treatment (*T. urticae* with and without competitor and *T. ludeni* with and without competitor). The random effects were time and the different island replicates within their experimental block. Model selection, Wald $\chi^2$ test, and pairwise comparisons were performed as explained above.

### Performance of T. urticae

Because we were interested in the magnitude of the differences in performance due to the presence of *T. ludeni*, we did a further analysis including also the control population of *T. urticae* on bean and the populations of *T. urticae* on cucumber without *T. ludeni*. We investigated the fecundity as a function of the three different treatments (control of *T. urticae* on bean, the populations of *T. urticae* on cucumber without *T. ludeni*, and those with *T. ludeni*) and time (categorical variable; 2, 4, 6, 8, or 10 months), and their interactions. The replicate islands were treated as random effects and were nested within the two experimental blocks. Model selection, Wald $\chi^2$ test of the maximal model, and pairwise comparisons of the variables for the best-fitting model were performed as explained above. Only results of the best-fitting model are presented below (summary statistics for full models are provided in the electronic Table S7).

The estimates provided in the tables are the raw and untransformed estimates for the fixed effects of the final models (negative binomial distribution). All analyses were performed in R (version 3.6.0) with glmmTMB version 0.2.3 (*Brooks et al., 2017*), MuMIn version 1.43.6 (*Bartoń, 2019*), and emmeans version 1.3.5.1 (*Lenth, 2019*).

**Table 1 Model selection.** Overview of the best models based on the lowest AICc with an AICc weight of at least 0.100. Abbreviations of fixed variables in maximal model: fecundity (fec.), dens. Tu (initial density *T. urticae*), dens. Tl (initial density *T. ludeni*), t (time), Tu comp./no comp. (*T. urticae* with competition/without competition), Tl no comp. (*T. ludeni* without competition), and treat. (treatment).

| Model | df | LogLik | AICc | Δ AICc | AICc weight |
|---|---|---|---|---|---|
| **The dynamics and performance of the ghost competitor** | | | | | |
| Max. model: fec. ~ plant species × mite species (df = 5) | | | | | |
| Plant species × mite species | 5 | −415.037 | 840.7 | 0.00 | 0.948 |
| **Signature of the ghost competitor on performance of *T. urticae*** | | | | | |
| Fecundity assessed on bean | | | | | |
| Max. model: fec. ~ t + dens Tu + dens Tl + t : dens. Tu + t : dens. Tl + (1\|block/island) (df = 18) | | | | | |
| No fixed effects | 4 | −768.410 | 1545.1 | 0.00 | 0.237 |
| Time | 8 | −764.312 | 1545.5 | 0.45 | 0.190 |
| Initial density *T. urticae* | 5 | −767.785 | 1545.9 | 0.87 | 0.153 |
| Fecundity assessed on cucumber | | | | | |
| Max. model: fec. ~ t + dens. Tu + dens. Tl + t : dens. Tu + t : dens. Tl + (1\|island) (df = 17) | | | | | |
| Initial density *T. ludeni* | 4 | −594.802 | 1197.9 | 0.00 | 0.484 |
| Initial density *T. urticae* + init. dens. *Tl* | 5 | −594.602 | 1199.6 | 1.74 | 0.203 |
| Initial density *T. urticae* | 4 | −596.243 | 1200.8 | 2.88 | 0.115 |
| Demography from plateau phase | | | | | |
| Max. model: dens. ~ treat. (*Tu* comp./no comp., *Tl* no comp.) + (1\|t) + (1\|block/island) (df = 7) | | | | | |
| Treatment | 7 | −1,185.931 | 2386.2 | 0.00 | 1 |
| Performance of *T. urticae* | | | | | |
| Max. model: fec. ~ treat. (*Tu* control/*Tu* comp./*Tu* no comp.) × t + (1\|block/island) (df = 18) | | | | | |
| Treatment | 6 | −1,846.641 | 3705.5 | 0.00 | 0.751 |
| No fixed effects | 4 | −1,849.890 | 3707.9 | 2.40 | 0.226 |

# RESULTS

## The dynamics and performance of the ghost competitor

In the competition treatments *T. ludeni* went extinct after approximately 2 months. While *T. ludeni* was able to maintain a population on cucumber in the absence of a competing species (12.3 individuals/cm$^2$), it reached a significantly lower density than *T. urticae* (19.8 individuals/cm$^2$, *t* ratio = 7.299 and *p* < 0.0001 for *T. urticae* under ghost competition; 20.5 individuals/cm$^2$, *t* ratio = 7.756 and *p* < 0.0001 for *T. urticae* without competitor). Although the significantly lower performance of *T. ludeni* during the fecundity assessment could reflect difficulties dealing with the host plant, competitor *T. ludeni* was able to maintain a population on cucumber in the absence of *T. urticae*. This indicates that the presence of *T. urticae* hindered the survival of *T. ludeni* (Figs. 2A, 2B; Tables 1–3).

The fecundity assessments on the initial and novel host plant with mites from the control populations of *T. urticae* and *T. ludeni* (which had been maintained on bean plants and had never been on cucumber before) showed that *T. urticae* had a significantly lower fecundity on cucumber than on bean (27.9 vs. 48.0 eggs after 6 days; *t* ratio = −3.629 and *p* = 0.0025), while there was no difference in the performance of *T. ludeni* on bean or

**Table 2 Chi-square statistics for the maximal models before model selection.** The results for the Wald Chi-square tests are presented for the maximal models. The number of asterisks determines the level of significance: one asterisk denotes a *p* value lower than 0.5, two asterisks lower than 0.01 and three asterisks lower than 0.001.

|  | Independent variables | Chisq | Df | Pr(>Chisq) | |
|---|---|---|---|---|---|
| The dynamics and performance of the ghost competitor | | | | | |
|  | Plant species | 5.1851 | 1 | 0.0228 | * |
|  | Mite species | 56.1019 | 1 | 6.881E−14 | *** |
|  | Plant species : mite species | 9.0064 | 1 | 0.0027 | ** |
| Signature of the ghost competitor on performance of *T. urticae* | | | | | |
| Fecundity on bean | Time | 8.4890 | 4 | 0.0752 | |
|  | Initial density *T. ludeni* | 0.8919 | 1 | 0.3450 | |
|  | Initial density *T. urticae* | 1.4948 | 1 | 0.2215 | |
|  | Time : init. dens. *Tl* | 1.3088 | 4 | 0.8599 | |
|  | Time : init. dens. *Tu* | 4.5696 | 4 | 0.3344 | |
| Fecundity on cucumber | Time | 4.6468 | 4 | 0.3255 | |
|  | Initial density *T. ludeni* | 4.1050 | 1 | 0.0428 | * |
|  | Initial density *T. urticae* | 0.6019 | 1 | 0.4379 | |
|  | Time : init. dens. *Tl* | 1.7385 | 4 | 0.7837 | |
|  | Time : init. dens. *Tu* | 4.0678 | 4 | 0.3969 | |
| Demography (from plateau phase) | Treatment | 74.1960 | 2 | <2.2E−16 | *** |
| Performance of *T. urticae* | | | | | |
|  | Treatment | 9.8340 | 2 | 0.0073 | ** |
|  | Time | 0.8189 | 4 | 0.9359 | |
|  | Treatment : time | 7.9751 | 8 | 0.4359 | |

cucumber (13.3 vs. 16.2 eggs after 6 days; *t* ratio = −1.012 and *p* = 0.7426). The fecundity assessments also showed that *T. ludeni* laid significantly fewer eggs than *T. urticae* on both bean (13.3 vs. 48.0 eggs after 6 days; *t* ratio = −7.463 and *p* < 0.0001) and cucumber (16.2 vs. 27.9 eggs after 6 days; *t* ratio = −6.177 and *p* < 0.0001). This suggests that the fecundity of *T. ludeni* on the novel host was already lower than that of *T. urticae* at the onset of the experiment (Fig. 2C; Tables 1–4).

## Signature of the ghost competitor on performance of *T. urticae*

Throughout the evolutionary experiment, we measured the densities of the populations of both spider mite species. During the 1st month, the ghost competitor (*T. ludeni*) was still present and the early competitive pressure calculated during the 1st month gave an indication of the pressure exerted by the ghost species on *T. urticae*. We found that the early competitive pressure of the ghost competitor positively affected the evolved individual fecundity of *T. urticae* on the novel host plant, cucumber, during the fecundity assessment on cucumber (Fig. 3B; *z* value = 2.33 and *p* = 0.0199). The evolved fecundity of individuals from populations under lower competitive pressures was lower than for individuals from populations under a higher early competitive pressure of *T. ludeni* (26.8 eggs after 6 days for *T. urticae* under a density of one individual of *T. ludeni* per cm$^2$

**Table 3 Pairwise comparisons adjusted for multiple comparisons (Tukey method).** The estimates provided in the table are the raw and untransformed estimates (negative binomial distribution). The estimates are the differences in fecundity for (A) and (C) and in density for (B). The number of asterisks determines the level of significance: one asterisk denotes a $p$ value lower than 0.5, two asterisks lower than 0.01 and three asterisks lower than 0.001.

| Contrast | Estimate | SE | df | $t$ ratio | $p$ Value | |
|---|---|---|---|---|---|---|
| **A. The dynamics and performance of the ghost competitor** | | | | | | |
| Comparison control population *Tu* and *Tl* on bean and cucumber (at first measured time point) | | | | | | |
| *T. ludeni* (bean) – *T. ludeni* (cucumber) | −0.199 | 0.197 | 97 | −1.012 | 0.7426 | |
| *T. ludeni* (bean) – *T. urticae* (bean) | −1.284 | 0.172 | 97 | −7.463 | <0.0001 | *** |
| *T. ludeni* (bean) – *T. urticae* (cucumber) | −0.742 | 0.173 | 97 | −4.279 | 0.0003 | *** |
| *T. ludeni* (cucumber) – *T. urticae* (bean) | −1.085 | 0.176 | 97 | −6.177 | <0.0001 | *** |
| *T. ludeni* (cucumber) – *T. urticae* (cucumber) | −0.543 | 0.177 | 97 | −3.068 | 0.0146 | * |
| *T. urticae* (bean) – *T. urticae* (cucumber) | 0.542 | 0.149 | 97 | 3.629 | 0.0025 | ** |
| **B. Signature of the ghost competitor on performance of *T. urticae*** | | | | | | |
| Influence of interspecific competitor on demography (from plateau phase) | | | | | | |
| *T. urticae* (comp.) – *T. urticae* (no comp.) | −0.034 | 0.063 | 325 | −0.544 | 0.8498 | |
| *T. urticae* (comp.) – *T. ludeni* (no comp.) | 0.475 | 0.065 | 325 | 7.299 | <0.0001 | *** |
| *T. urticae* (no comp.) – *T. ludeni* (no comp.) | 0.510 | 0.066 | 325 | 7.756 | <0.0001 | *** |
| **C. Performance of *T. urticae*** | | | | | | |
| Investigate local adaptation (pooled across time points) | | | | | | |
| *T. urticae* (no comp.) – *T. urticae* (comp.) | 0.1119 | 0.0515 | 453 | 2.174 | 0.0767 | |
| *T. urticae* (no comp.) – *T. urticae* (control) | 0.1605 | 0.0516 | 453 | 3.110 | 0.0056 | ** |
| *T. urticae* (comp.) – *T. urticae* (control) | 0.0486 | 0.0516 | 453 | 0.941 | 0.6144 | |

compared to 33.1 eggs with four individuals of *T. ludeni* per cm$^2$). This effect emerged from the start of the experiment and was independent of time as the best-fitting model did not include time.

An extra analysis was performed to investigate whether excluding the 1st months from the dataset still gave the same results (electronic Table S8). The early competitive pressure of the ghost competitor was still included as the only explanatory variable in the best-fitting model when excluding the 2nd month, both the 2nd and 4th months, and the 2nd, 4th and 6th months. When we considered each time point by itself, this result was not found, which is likely a lack of power.

The early competitive pressure of the ghost competitor did not explain the fecundity assessed on the original host plant, bean (not included in the best-fitting model). Also, the density of *T. urticae* itself or the total initial density was not related to performance during the fecundity assessments on both bean and cucumber leaf discs (Tables 1–5).

The early competitive pressure of the ghost competitor did not influence the demography of *T. urticae*. All *T. urticae* populations reached similar densities during the plateau phase regardless of the initial presence of the ghost competitor (19.8 and 20.5 ind./cm$^2$ respectively; $t$ ratio = −0.666 and $p$ = 0.7833). At the start, the density of *T. urticae* without competition was temporarily higher than the populations of *T. urticae*

**Table 4 Summary of the final best-fitting GLMM explaining reproductive performance.** The values provided in the table are the raw and untransformed estimates due to the negative binomial distribution in the model. The number of asterisks determines the level of significance: one asterisk denotes a *p* value lower than 0.5, two asterisks lower than 0.01 and three asterisks lower than 0.001.

| | Estimate | SE | z value | p value | |
|---|---|---|---|---|---|
| The dynamics and performance of the ghost (fecundity at first measured time point) | | | | | |
| (Intercept) (*T. ludeni* on bean) | 2.5867 | 0.1367 | 18.92 | <2E−16 | *** |
| Cucumber | 0.1990 | 0.1965 | 1.01 | 0.3114 | |
| *T. urticae* | 1.2838 | 0.1720 | 7.46 | 8.47E−14 | *** |
| Cucumber : *T. urticae* | −0.7407 | 0.2468 | −3.00 | 0.0027 | ** |
| Signature of the ghost competitor on performance of *T. urticae* | | | | | |
| Fecundity assessed on bean (pooled across time points) | | | | | |
| (Intercept) | 3.7098 | 0.0899 | 41.27 | <2E−16 | |
| Fecundity assessed on cucumber (pooled across time points) | | | | | |
| (Intercept) | 3.2209 | 0.0883 | 36.47 | <2E−16 | *** |
| Initial density *T. ludeni* | 0.0693 | 0.0298 | 2.33 | 0.0199 | * |
| Density after plateau phase | | | | | |
| (Intercept) (*T. ludeni* without comp.) | 2.5095 | 0.0539 | 46.55 | <2E−16 | *** |
| *T. urticae* with comp. | 0.4751 | 0.0651 | 7.30 | 2.91E−13 | *** |
| *T. urticae* without comp. | 0.5095 | 0.0657 | 7.76 | 8.79E−15 | *** |
| Performance of *T. urticae* (fecundity pooled across time points) | | | | | |
| (Intercept) *T. urticae* without comp. | 3.5241 | 0.0395 | 89.23 | <2E−16 | *** |
| *T. urticae* under comp. | −0.1119 | 0.0515 | −2.17 | 0.0297 | * |
| *T. urticae* control | −0.1605 | 0.0516 | −3.11 | 0.0019 | ** |

with *T. ludeni* present (Fig. 2A), which is most likely due to the differences in starting densities of *T. urticae* between both treatments.

## Performance of *T. urticae*

We compared the performance of mites from a control population that was maintained on bean plants with mites adapting to cucumber where in both cases the interspecific competitor was replaced by conspecifics. The evolved individual fecundity of the control population was significantly lower for the assessment on cucumber than the evolved individual fecundity of populations grown on cucumber (28.9 vs. 33.9 eggs after 6 days; *t* ratio = −3.110 and *p* = 0.0056), which suggests local adaptation to the novel host plant for the latter group (Fig. 3B; Tables 1–4).

## DISCUSSION

The process of genetic adaptation to novel environmental conditions is typically studied and understood from the perspective of the available genetic variation and selection pressures as imposed by the environment. Because competing species are an intrinsic part of novel experienced environmental conditions, they are known to mediate sometimes complex evolutionary processes. Here we provide empirical evidence that initial

**Table 5 Model selection (A) and Wald $\chi^2$ test (B) for the influence of total initial density on fecundity.** Overview of the best models based on the lowest AICc with an AICc weight of at least 0.100.

| (A) Model | df | LogLik | AICc | ΔAICc | AICc weight |
|---|---|---|---|---|---|
| Fecundity assessed on bean – max. model: fecundity ~ time + total initial density + time : total initial density + (1\|block/island) | | | | | |
| No fixed effects | 4 | −768.410 | 1,545.1 | 0.00 | 0.293 |
| Total initial density | 5 | −767.371 | 1,545.1 | 0.04 | 0.287 |
| Time | 8 | −764.312 | 1,545.5 | 0.45 | 0.235 |
| Time + total initial density | 9 | −763.522 | 1,546.2 | 1.10 | 0.170 |
| Fecundity assessed on cucumber – max. model: fecundity ~ time + total initial density + time : total initial density + (1\|island) | | | | | |
| No fixed effects | 3 | −597.445 | 1,201.1 | 0.00 | 0.617 |
| Total initial density | 4 | −597.132 | 1,202.5 | 1.48 | 0.294 |

| (B) | Independent variables | Chisq | Df | Pr(>Chisq) |
|---|---|---|---|---|
| Fecundity on bean | Time | 8.3414 | 4 | 0.0798 |
| | Total initial density | 1.5149 | 1 | 0.2184 |
| | Time : total init. dens. | 4.4325 | 4 | 0.3506 |
| Fecundity on cucumber | Time | 4.2049 | 4 | 0.3790 |
| | Total initial density | 0.7819 | 1 | 0.3766 |
| | Time : total init. dens. | 3.8180 | 4 | 0.4312 |

competition between two species can have a long-lasting effect on their performance in a novel environment.

The unintentional rapid extinction of *T. ludeni* seems a logical consequence of the higher attained fecundity of *T. urticae* on the novel host already at the onset of the experiment (Fig. 2C). This higher fecundity and hence higher growth rate increased the chance for better establishment or recovery after disturbance (*Turcotte, Reznick & Hare, 2011, 2013*). Also, populations from *T. urticae* were at a higher density at the plateau phase than populations from *T. ludeni* (Fig. 2B). The density of *T. urticae* on the measured surface was almost fifty percent more than the density of *T. ludeni* when grown alone. This suggests that *T. urticae* has a higher resource efficiency than *T. ludeni*, which could for instance arise from evolved detoxification mechanisms as often found between herbivores and their hosts (*Després, David & Gallet, 2007*; *Dermauw et al., 2018*). The population size at the plateau phase might not only be due to a signature of ecological dynamics, but may also be a consequence of adaptation itself, because the carrying capacity may be affected by organismal traits. In this case, our results suggest selection for growth rate but not for carrying capacity due to ghost competition. After 1 month the density of populations of *T. urticae* without heterospecific competition was higher than that of populations with heterospecific competition, but this difference vanished together with the extinction of the competitor. This probably means that the ghost competitor (when still present) decreased the available resources resulting in a lower population size for *T. urticae* (Fig. 2A). Another explanation is a delayed growth due to the lower initial population size in the experiment.

We have shown that the higher the competitive pressure of the ghost competitor (as measured by its average density in the 1st month), the higher the fecundity of *T. urticae* was on the novel host plant (Fig. 3B). We speculate that a higher selection pressure was exerted under a higher early competitive pressure of the ghost competitor, which eventually led to an increase in fecundity of the focal species. It is known that the competitor, *T. ludeni*, can down-regulate plant defences (*Godinho et al., 2016*), but this cannot explain the correlation between its density and the fecundity of the focal species even long after the ghost competitor went extinct, because plants were refreshed weekly.

Furthermore, we compared the evolved individual performance of mites from a control population (maintained on the initial host plant) with mites adapting to cucumber with or without the ghost competitor. We found that individuals from populations of *T. urticae* grown on cucumber plants without *T. ludeni* reached a higher evolved fecundity on the novel host plant than individuals from the control population (maintained on the initial host plant), implying local adaptation to the novel host (Fig. 3B). This difference in evolved individual fecundity with the control population was not found for individuals from the populations that were initially under competition with *T. ludeni* when considering all these populations as a single group (so not splitting them according to initial population size of *T. ludeni*). However, the fecundity of this group under interspecific competition was also not significantly different from the adapted *T. urticae* populations without interspecific competition, indicating that this group was intermediate between the adapted group without interspecific competition and the control population.

We explain this by the various degrees of early competitive pressure in this group under heterospecific competition, the main result of our study. We suggest that not finding a difference in fecundity with the control population may be due to an initial trade-off between performance on the novel host and the ability to compete with heterospecifics (*Siepielski et al., 2016*). It is possible that the populations under competition initially adapted to the competitor and that this lowered the amount of standing genetic variation necessary for adaptation to the novel host. A likely alternative explanation for these various degrees of early competitive pressure is variation in drift effects which is more probable in the populations under heterospecific competition because of the lower initial population size per species. Under a scenario of strong drift effects we would expect large differences in performance between replicates with and without *T. ludeni* (and thus with high or low initial population size for *T. urticae*), but they were quite similar (Fig. S1), so strong drift can be largely discarded as a driver of the observed evolutionary dynamics.

Individuals from populations where half of the population was replaced with heterospecifics could not adapt as well to the novel host plants as the populations consisting of only conspecifics. This could be due to a higher selection pressure from conspecifics compared to heterospecifics which may in turn be due to the larger initial population sizes or due to a larger niche overlap (*Bolnick, 2001*; *Svanbäck & Bolnick, 2007*).

The history of species in a community can have an impact on interspecific interactions (*Fukami, 2015*). The magnitudes of such historical contingencies do, however, strongly differ among species and environments (*Vannette & Fukami, 2014*). Differences in historical contingency may explain why some populations experience radiations, whereas others from

the same clade do not achieve this under seemingly similar conditions (*Seehausen, 2007*). Our results suggest that increased interspecific competition leads to higher selection pressures and thus improved performance (Fig. 3B). Our results coincide in this respect with other empirical work demonstrating that increased competition with heterospecifics increased local adaptation in bunchgrasses (*Rice & Knapp, 2008*). Similarly, intraguild predation between lizard species increased the selection pressure and led to strong divergence in morphological adaptation as associated with niche specialisation (*Stuart et al., 2014*).

Nevertheless, we have to be careful with generalising our results. First, we chose small populations sizes as they are more biologically relevant, but this may limit adaptation and establishment (*Del Castillo et al., 2011*; *Yates & Fraser, 2014*). We also used populations that have been maintained in the lab for many generations, probably leading to a decrease in genetic variation compared to wild populations. The problems we encountered in creating isofemale lines for *T. ludeni* could be an indication of inbreeding depression. However, we are confident that our results are robust as we could still provide evidence for local adaptation in the populations of *T. urticae* without competitor (Fig. 3B). This suggests that the initial amount of genetic variation did not limit *T. urticae* in our study.

Second, it is impossible to add a competitor without changing total population sizes, population densities, or island sizes; all of these affect genetic variation and drift (*Del Castillo et al., 2011*; *Alzate, Etienne & Bonte, 2019*). As it is known that larger populations usually contain more genetic variation, we chose to standardise this by means of isofemale lines, knowing that this might create differences in drift among treatments. One way to better disentangle the effects of drift from those of selection with our small population size would have been to increase the number of replicates which was difficult for logistical reasons.

Third, our experimental design is not strictly suitable to assess adaptation in the interspecific competition treatment, as we did not keep a heterospecific control population on bean (again for logistical reasons). Hence, we cannot disentangle the effect of changes in fecundity due to competition (independent of the novel environment) from the effect of competition on adaptation to the novel environment. Although this means that we cannot detect adaptation in the treatment under ghost competition, we did find a positive influence of the density of the ghost competitor on fecundity, meaning that early competitive pressures substantially matter and providing evidence for eco-evolutionary dynamics.

## CONCLUSIONS

In conclusion, we have shown the importance of early selection pressures such as ghost competition. Even when one species becomes extinct, the competition signature continues to affect the adaptation process of the successful species.

## ACKNOWLEDGEMENTS

We thank Viki Vandomme, Angelica Alcantara, Pieter Vantieghem, Katrien Van Petegem, Stefano Masier, Matti Pisman, Mike Creutz, Hilde De Nil and Johan Bisschop for helping during the research experiments, and to Sarah Magalhães for providing the strains of

*T. ludeni*. We thank the Terrestrial Ecology department of Ghent University and the Centre for Ecology, Evolution and Environmental Changes of the University of Lisbon for the spider mite populations.

### Funding

This work was supported by the NWO (No. 865.13.00), the Special Research Fund (BOF) of Ghent University, the Ubbo Emmius sandwich programme of the University of Groningen, the FWO network EVENET (No. W0.003.16N) and the FWO (No. G018017N). The funders had no role in study design, data collection and analysis, decision to publish, or preparation of the manuscript.

### Grant Disclosures

The following grant information was disclosed by the authors:
NWO: 865.13.00.
Special Research Fund (BOF) of Ghent University, the Ubbo Emmius sandwich programme of the University of Groningen, the FWO network EVENET: W0.003.16N.
FWO: G018017N.

### Competing Interests

The authors declare that they have no competing interests.

### Author Contributions

- Karen Bisschop conceived and designed the experiments, performed the experiments, analysed the data, prepared figures and/or tables, authored or reviewed drafts of the paper, and approved the final draft.
- Frederik Mortier conceived and designed the experiments, performed the experiments, authored or reviewed drafts of the paper, and approved the final draft.
- Dries Bonte conceived and designed the experiments, authored or reviewed drafts of the paper, and approved the final draft.
- Rampal S. Etienne conceived and designed the experiments, authored or reviewed drafts of the paper, and approved the final draft.

### Data Availability

The data and R script are available at: Bisschop, Karen; Mortier, Frederik; Bonte, Dries; Etienne, Rampal S., 2020, "Replication Data for: Performance in a novel environment subject to ghost competition", https://hdl.handle.net/10411/VETYPE, DataverseNL, V1.

### Supplemental Information

Supplemental information for this article can be found online at http://dx.doi.org/10.7717/peerj.8931#supplemental-information.

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
