# Peer review of "Performance in a novel environment subject to ghost competition"

_PeerJ, doi:10.7717/peerj.8931_

## Round 0.1 · original submission · Major Revisions

Three reviewers have now reviewed your manuscript. All three reviewers - as well as myself - find the questions you're asking to be interesting and your experimental approach to be well-done. That being said, the reviewers identified a number of issues with this manuscript, particularly with respect to whether the narrative matches what can actually be said about the findings, that necessitate substantial revisions.

In large part, these issues pertain to interpretation and discussion of the results. In some cases, these issues can be addressed by carefully working on the introduction and discussion. Reviewers 1 and 3 highlighted a number of these instances which likely require substantial revision. But as Reviewer 2 notes, this study wasn't designed to address the issue that is the focus of the manuscript and so the revision should more carefully address the original aim of the study but also what we can learn from the outcomes - whether these outcomes were intended or not. The reviewers also highlighted a number of instances where understanding parts of the experimental design were complicated to understand.

Critically, Reviewer 3 highlights a number of statistical issues that may be influencing the findings and the downstream interpretation of the results. The analyses are important here because the reviewers noted several areas of the manuscript where the interpretation of the study either do not match the outcome of the analyses or where the analyses themselves were insufficient.

I very much look forward to a revised version of this manuscript.

Reviewer 1 ·

Basic reporting

This manuscript is well-written and professional. Most of the text is easy to comprehend. Though the experimental design is very good, it is also very complicated. I had to read this several times to fully understand exactly what was done here. The methods would benefit from a diagram showing the different evolutionary environments and the current ecological environments.
Some of the references in the text are not listed in the references at the end.

Experimental design

The manuscript proposes an interesting question to be tested, namely, do competitors (different mite species) affect selection to a novel environment (in this case, the environment is the plant on which the mites grow). Moreover, they serendipitously are able to examine the effect of ghost species...those that are in the community for some time and affect community or population (or evolutionary) dynamics, even though they are not a part of the final community. The design is a good one for testing the question posed. As stated above, the design is complicated and would benefit from a summary diagram.

-There is one statistical table presented, though other tests were also performed. These should be included in the main text as well. Perhaps it is because violin plots were used, but the statistical results do not seem to support what is shown in the graphs. I think violin plots may not be the most useful for demonstrating differences between treatments, but this also makes me question many aspects of the statistical design. Were Blocks included as a factor in all analyses? It seems as though they were in some, but not others, but this may just need clarification. The same can be said of Day. I could not tell in which analyses Day was included as a factor, or fecundity was averaged across days. This deserves some clarification or reanalysis.

Validity of the findings

Although I very much like the ideas being tested here and am excited that somebody is testing the effect of ghost species on evolution (seriously, I really like this idea!), I completely disagree with the conclusions.
First, the authors conclude that the presence of an interspecific competitor affects evolution in response to a novel environment (cucumber plants), but I do not see any evidence for that. They seem to base this conclusion on Fig. 2b, where they claim the density of competitors in the selection environment affects fecundity in the novel ecological environment, but that stats do not support this conclusion. There is no significant difference between the 0 interspecific competitors and any of the other densities. The only significant difference is between zero competitors and the plot on the right, though despite a lot of time trying to figure it out, I can't figure out what treatment this plot represents. It doesn't appear to be on the same x-axis as the other violin plots, or if it is, I do not understand how the highest density of competitors can be in the "no comp" treatment.
Second, the authors conclude that the "ghost" species has an effect on evolution, but I disagree. Any competition experiment that tracks growth over time would expect that when one competitor is removed (as one mite species here goes extinct after two months), the remaining species would increase in abundance, and that this increase in abundance will take some time. The same happen here. The ghost species has a temporary effect that, when removed, the community takes some time to recover from. I would not expect anything different. I would only consider this an effective ghost species if the effect were permanent and stable. At least that is my understanding of ghost species from the terHorst, Burns, and Miller paper. What is shown here is just a transient effect of competition.
Finally, there is a lot of focus here on the possible effects of genetic drift, though I am not sure why this is given so much attention. If one finds a difference in selection environments, with replicates within those environments, that indicates that selection had an effect that must have been much stronger than any effect of genetic drift. The authors are to be applauded for considering drift and the effect of abundance, but given that it has so little effect, I do not think that they need to spend much space discussing it. In that sense, Fig. 3 is not necessary.

Reviewer 2 ·

Basic reporting

I really enjoyed reading this manuscript. The authors communicated in clear unambiguous, professional English. In terms of the amount of background information, I feel that the authors have done a sufficient job with what they have reported and discussed, but ultimately that they ought to reframe their manuscript, and so will have to find the relevant literature for that. The figures are nice and easy to follow.

Experimental design

I think the results of this study are very interesting, and while I appreciate that the authors attempted to make something out of what others may have considered a failed experiment, I think that the way they did this is not great and that the manuscript requires some reframing.

1. the experiment was not initially designed around testing the effect of ghost competition. While this may be an explanation for their results, I feel that it is disingenuous to use these results a test for that. I would reframe this manuscript to test their initial hypothesis, then in the discussion suggest ghost competition as one possible reason for observing the results that they did.

2. The authors used inbred lines of mites, and did an evolutionary experiment with them. Was there enough genetic variation in the population to allow for evolution? If the T. urticae population evolved initially, perhaps there wasn’t enough genetic variability remaining for it to evolve again after the extinction of T. ludeni, and that the long term results seen (25 generations) are due to this, rather than lasting effects of ghost competition.

Validity of the findings

This is covered under experimental design.

Additional comments

Overall, I did really enjoy reading this paper and found it interesting. In addition to my comments above, I have some minor comments that I refer to by line numbers:

Line 30 - populations are the unit for evolution, not species. I think this sentence would be improved by rewording.

Line 26 - the word “even” does not need to be italicized. But also, could be removed entirely and the sentence would be better.

Line 89 - I don’t think it’s necessary to say that it’s an underestimated process (compared to what?). This sentence could be rewritten as “We therefore suggest that ghost competition may lead to differences in long-term local adaptation.”

Lines 213-214 - Repetitive text?

Line 393 - replace “are affecting” with “affect”

Line 407 - replace “we did find” with “we found”

Reviewer 3 ·

Basic reporting

The English is decent, but sometimes feels awkwardly worded (for example, line 158, 160-166; see “general comments”). Several aspects of the experiment were not well explained, notably what “host islands” were (used but not introduced on line 126), and the distinction between replicates and host islands (was there one?). Experimental design involved insects on two different host plant species, with fecundity evaluated on two different host plant species, and it was sometimes difficult to tell which of those were being compared (e.g. insects from the same host plant species evaluated on two host plant species, or individuals from each host plant species evaluated on different host plant species). This is inherently a difficult distinction to get across, and some careful rewording is in order in some cases (see general comments for examples).

Figure 1B has erroneous confidence intervals, and letter groupings are not explained. This may also be the case for figure 2, which appears to use the same code. The black lines of figure 3 are identical to those used to represent 95% CI in other figures in this paper, yet instead represents range of the data in this case (which we can already see as the raw data is all present in this figure); this may confuse readers.

Experimental design

The research question is a good one – how does the presence of a heterospecific population influence the evolutionary trajectory of a population. The experiment was carried out across 10 months, and tracked performance of populations using the fecundity of the standardized grandchildren of 5 randomly sampled individuals per time point. The experimental design introduces a number of complicating factors, including (a) differing population sizes between intra- and inter-specific competition treatments, (b) regular “disturbance events” in which new plants were added and old ones were removed, necessitating colonization, (c) the use of fecundity in the absence of competition as a proxy for fitness (typically we think that performance at high densities determines competitive abilities), and (d) using a sample of 5 individuals at a time point as a proxy for the total population, when each replicate started with 13 distinct genotypes (isofemale lines). I note that some of these factors are necessitated by the system and the overall type of experiment (it is impossible to simultaneously control population size, patch size, and overall density), and experiments in ecology are often necessarily messy.

The statistical analysis leaves me with several concerns. First, the model choices seem prone to overfitting – in particular, treating the 10 months as a categorical predictor, and then allowing interactions between that and other factors in addition to nested random effects leads to a substantial number of estimated coefficients. The authors report running into issues with singular fits, which I also experienced while fitting simpler models to test their observation that T. urticae evolved to have higher fecundity on cucumbers when early T. ludeni populations were higher. One explanation for that would be that, with only 8 replicates for this treatment, early T. ludeni population size might largely be serving as a stand-in for island identity. Second, effect sizes are not reported, and the values in Table 1 are not in the raw units despite their description. Third, the statistical analyses are exploratory – this is fine, but p-values lose their meaning when calculated after model selection. In essence, by testing many potential predictors (for example, “initial” densities of each herbivore species crossed by host plant species) and focusing only on those predictors that model selection identifies as important, we risk capturing spurious relationships and elevating them to statistical significance.

Trajectories of population densities through time across ~25 generations are fit using a quadratic model (line 241) – this is not something we would expect biologically or phenomenologically, and while it is not the emphasis of this paper, I do not think this is the correct approach to take. It would be more appropriate to (a) use existing models for population growth (for example, logistic growth), or (b) use flexible methods like penalized spline-fitting. I would focus on (b), as (a) requires choosing the correct functional form for population growth in this system. On the other hand, (a) can provide estimates of biologically interesting quantities like carrying capacity (although note that with the weekly colonization/plant removal process, interpretation of model parameters is complicated). By fitting a quadratic equation, you are implicitly assuming that growth itself is quadratic through time. One of the issues of quadratic equations is that they have a constrained curvature - for a trajectory to increase more steeply early than it does late (e.g. density-dependent growth behavior), the model necessarily will include a negative quadratic coefficient, which will ensure that for a late enough time, the population size will go into the negatives. We see the start of this with the projection of T. ludeni (no comp) at the end of the study in Fig. 1A, and the reverse problem with T. ludeni (comp) right as the projections are cut off (the line is starting to point upward, and would trend towards infinity with increasing time).

More generally, if the question is the long-term effect of “ghost” competition on population fecundity, it seems that the most important measure is the population behavior at the end of the experiment, not midway through. Further, since all replicates within a treatment began with (hypothetically) identical starting populations, the relationship between early population sizes and later fecundity are correlative, and may reflect some latent variable(s) determining island quality.

Validity of the findings

The data is provided but lacks the metadata to explain what the various columns are. Two column in the data frames include the filepath of the original photos on the author’s computer, which do not appear to have any use to readers. These and any other truly extraneous columns (X, which appears to be row numbers from a past data frame?) could be removed.

As mentioned above, I have concerns with the statistical methods, and the results that derive from those do not feel statistically sound. In some cases, the authors provide statements that do not appear to be supported by the data (e.g. on line 359 the others suggest that selection pressure of conspecifics is higher than heterospecifics in this experiment, but I am unclear the evidence; see General Comments for any other cases). In several places the authors present predictors as having no effect when they mean that the predictor was not included in the best-fitting model – a subtle distinction, but one worth making (e.g. line 292-293, the best-fitting model did not include time, but this does not mean that the effect is stable through time, see General Comments for a few more examples). The key results of this study are several layers removed from the claims of their conclusion – the impact of the ghost species was detected when pooling all time points, but was not detected (at least not when I tried with their data) when looking at the latest time points.

Additional comments

Throughout the paper, the authors use “initial” densities to refer to population densities averaged (or summed?) across the first two months. I would suggest using a different word, as the actual initial population sizes were the 52 individuals at the start of the experiment.

Analysis of the effect of competition on population density seem to ignore the fact that each species started with half as many individuals in the competition experiments (e.g. line 297). We would expect that with half the initial population size, subsequent population censuses should be quite different (at least until density dependence evens things out).

This is a system with repeated disturbance; in general there are expectations that disturbances may enable coexistence which otherwise isn’t possible. Colonization, while important, isn’t a fundamental part of much of coexistence. I would like to see some explanation for how you think this might/might not matter.

This paper is focused on explaining adaptation, suggesting that performance (in this case fecundities) can change through time. I think it would be appropriate to think carefully about when it’s appropriate to pool samples across time and when it’s not. Whatever choice the authors make, they should justify it to the reader.

MINOR COMMENTS

Line 35: Niche overlap is an idea that predates Bolnick 2001, and it might be appropriate to cite a more foundational paper as well.

Line 49: I’m surprised there aren’t earlier references than this; this seems like a specific case of coexistence theory.

Line 52, 62, possibly later: replace “coexisting” with “co-occurring”. Coexistence implies indefinite persistence of each species (see Siepielski and McPeek 2010, “On the evidence for species coexistence…”).

Line 56: this paragraph feels a bit jarring in that, as you found, interspecific competition can also simply cause extinction of one of the species. This extreme hindrance to local adaptation seems like it should be addressed rather than ignored.

Readers working in the plant-insect interactions world are likely to be thinking about plant induction, both as an effect that can change through time and something that can mediate interspecific interactions. You mention this in the discussion, but it seems like it would be valuable to address this sooner, perhaps in the introduction and/or methods.

Line 126: the term “islands” is used here and then throughout the paper, but never explained.
Line 134: I wouldn't call this natural movement dynamics. I think this section should be revised a bit. I'm on board for what was done, but the explanation felt lackluster. I would reorder to start by explaining the biology, and then walk through the plant addition/removal methods.

Line 152-155: Uncontrolled and undetected inconsistency in selective pressures between replicates isn't good in general, and especially when there are only 8 replicates total. It’s good to acknowledge it, but don’t refer to it as a benefit.

Line 158: “rather” is an informal word choice

Line 160-166: The wording here feels awkward. I would rephrase a bit to focus on the things you kept constant, then mention "this necessarily meant that the initial population size for each species was not the same across the different treatments; we recognize this may effect genetic drift and sampling effect".

Line 174: how does the every-two-week-sampling line up with the plant replacement scheme? How long did mites have to colonize the new plant before counting?

Lines 201-203: English is awkward

Where did the leaf disks in fecundity testing come from? Was anything done to account for induction in them?

Line 207: providing the error distribution without the response variable seems odd. Were you using the same distribution for population densities and fecundity measures?

Line 210: Typo – “parameterization”. Also, do you have a citation for this?

Line 219: If you’re using glmm, you were presumably including random effects. You should mention this here. In this case, because you have a total of 2 predictors, each of two factors, it would seem most appropriate to carry out an ANOVA comparison of the full model rather than carry out model selection.

Line 234: What does the result look like if you treat time as a continuous variable, though? If they are similar, report that. If not, report *that*. Breaking a continuous variable (time) into many factors is inherently concerning, and while your explanation makes sense, readers will be more comfortable knowing that you tried the simpler approach.

Line 241: This deserves its own (sub)section, as it appears to be the only analysis using population density instead of fecundity measures. This is also the only section in which time is treated as continuous instead of categorical.

Line 271: if T. ludeni reached significantly lower density than T. urticae, this could still reflect host plant problems (since this was not a choice experiment, and performance is not binary). We would expect to see a gradient in performance from good hosts to bad hosts to unusable hosts, much as we see with patch quality when looking at source-sink dynamics.

Line 298: these appear to be equilibrium densities, but we haven’t established this (and technically a quadratic model has no equilibrium density)

line 325-328 presupposes that the mites are at or near the carrying capacity – without a model that includes carrying capacity, this is hard to support.

unclear what 336-338 is referring to

Wording on lines 346-348 is a little unclear - make it obvious to the readers that in both cases you are talking about fecundity on the novel host plant. (Currently it can be misinterpreted as the T. urticae that evolved on the novel host plant have higher performance on that plant than the control population has on the control plant).

Lines 349-351: this is possible, but it could also be that mites have a tradeoff between their ability to perform on the plant vs their ability to compete with heterospecifics. It would help to know the extent to which mites compete by directly interacting, vs through resource competition.

sentence starting on line 354 is incomplete.

Line 359: What is our evidence for the selection pressure of conspecifics being higher than heterospecifics?

Sentence starting line 372 can be tightened up.

Wording on line 399-400 is a bit confusing. You mean that there were not control populations on beans? Use of "mixed" confuses things.

Fig 1:
• (A) What models were used to fit the data? It appears this was not done using the modeling of the methods section, and instead uses ggplot methods.
• (B) I thought we did not have controls on beans? This is performance on both hosts from control populations raised only on cucumber?
• (B) Confidence intervals for Fig 1B appear to be extremely wrong. Significance groups look good, but CIs in the figure do not at all match (a) significance groups, or (b) CIs when I calculate them. Sample code to make plot (which does not match Fig 1B) and calculate differences of groups (which matches paper results):
####
test=data.frame(total=control_data$Total, mp=control_data$MitePlant)
out.lm=lm(total ~ mp -1, data=test)
## note: this is ordering groups a bit differently - by mite and then by plant.
ci=confint(out.lm)
plot(1:4, out.lm$coefficients, ylim=c(0,100), pch=19)
segments(x0=1:4, y0=ci[,1], y1=ci[,2])
out.aov=aov(total ~ mp, data=test)
TukeyHSD(out.aov)
####


I found it interesting that the fecundity had a downward trend through time in both cases, with month as a continuous variable being a significant predictor of fecundity in the bean data.

Fig 2: by "minimal model" do you mean "best-fitting model"? Minimal could mean many things, including intercept-only.

Fig 3c: No data points for replicate 7? If so, remove from the figure as was done in fig 3a
What is the distinction between island and replicate?

Line 292-293: finding the best-fitting model did not include time does not mean that the effect is stable through time.

Lines 293-294: tell us the evidence for this.

CODE COMMENTS:

Note that the normal distribution appears to be almost as good a fit as the negative binomial, and the overdispersed negative binomial (which is the distribution used in model fitting) is not one that is evaluated during the distribution testing of "Dynamics and performance of the ghost competitor" section. lm(Total~TreatPlant*MiteSp, data=control_data) performs substantially better (AIC = 820) than the stmodel.0 (AIC = 840), suggesting that the simpler normal distribution may actually be more appropriate here.

In general, I would use and present the simplest test possible for the relationship in question. For example, the relationship between “initial” density of T ludeni and fecundity of T. urticae could be evaluated with a simple lmer model, of lmer(Total ~ initialdensL + (1 | CombinedName), data=cucumberdatacomp). In doing so, we find that there is a singularity in model fit, which may indicate that the predictive capabilities of initial density and island are overlapping.

---

## Round 0.2 · Minor Revisions

Thank you for responding to the reviewers' comments - your manuscript is certainly easier to follow now and is greatly improved.

The three reviewers have again assessed your manuscript. In general, all enjoyed the revised version. Even so, there are still some issues that require your attention. Reviewers 1 and 3 find the new version of your manuscript to still be challenging to read and comprehend. Both reviewers have provided a number of comments to help address this issue. I tend to agree with the reviewers in this case. While improved, the methodology, results, and discussion take a few rounds of reading before they can be understood. To be fair, the experiment is complex and further complicated by unexpected outcomes. Even so, taking some time to make your manuscript clearer will help.

Reviewer 3 again notes some statistical issues that require your consideration.

Additionally, I encourage you to more carefully consider Reviewer 2's critique as I believe it's an important concern with respect to honest reporting. As one approach, I suggest moving some text from the final paragraph of your Introduction into a new paragraph before you introduce the idea of ghost competition. Specifically, state what the initial goals of your experiment were, what the design was, and why you designed your experiment in a particular way to address these goals. Then briefly describe the issue that occurred in the experiment. End the paragraph by noting that the unexpected issues with the experiment provided serendipity such that you could capitalize on the novel situation to study ghost competition. Your final paragraph could then be the brief introduction to what ghost competition is with a final sentence about what your experiment concluded.

I look forward to receiving your revised manuscript.

Reviewer 1 ·

Basic reporting

I reviewed the first version of this manuscript. I continue to think that the idea of a ghost competitor affecting evolution is a really cool idea that can be tested with the data available here. I concur with the authors that they have demonstrated that the density of T. ludeni affects local adaptation of T. urticae. They have demonstrated that ghost species can alter evolution.

However, I really struggled with this manuscript for days to decide whether I agreed with the authors or not. Parts of the manuscript are really difficult to understand, particularly the Methods section. I would not consider this second version a major revision. Even though I was familiar with the story, it took me even longer to digest this manuscript than the first time. Many of the previous reviewers' comments were not incorporated, which would have greatly improved the clarity of the manuscript. I do not believe that this manuscript is yet ready for publication, as the really cool story they have to tell will be obscured by writing that is difficult to understand.

Experimental design

The experimental design is a fortunate accident, as they ended up able to test a question about ghost species that they had not intended to test. Nevertheless, the experimental design is useful for testing this question.
However, the description of the experimental design is quite poor. The new supplemental figure helps to a large degree. This should be moved into the main text to make the design easier for the reader to digest. I've read through the Methods several times and I could not find any mention of the Control treatment, where each species was continued to be raised on beans. This Control treatment is critical to interpreting the Results, but is not mentioned until then. How many replicates are there of the control treatment? How were they maintained? Were they treated the same as the experimental cucumber treatments?

Validity of the findings

Once I understood the findings, I agree with the authors. It just took me a VERY long time to interpret their design and results in order to determine that I agreed with them.

Additional comments

Additional comments:
-Fig. 1A would be easier to comprehend if the means and errors for each day were presented, rather than every data point. There are so many data points in the figure, that it makes it difficult to observe a pattern (some are even inside the legend). Also, each dot appears to have a light-colored box around it? This figure will also be impossible for the ~5% of the population who are color-blind to interpret. Means with connecting lines would fix this problem.
-The x-axis on Fig. 2 is quite strange. If the x-axis is meant to be the density of competitors, then the light green treatment (no competitors) should be on the left end of the axis, because there were 0 competitors. The yellow plot does not belong on this axis, as it is an entirely different control treatment in which competitor density was not manipulated. So that treatment cannot have a value on the x-axis.
-I do not see a lot of value in Fig. 3, and if it is to be included, should be in the supplemental material. I believe the intent is to show that drift is not important and that all replicates showed a similar result, but I believe that same information is already presented in Fig. 2 in the form of the error bars.
-Introduction, lines 30-33. I don't see how these two topics are related to this manuscript.
-Introduction, lines 36-37. I would argue that it is not generally expected that conspecific competition will have a bigger effect on evolution than heterospecific competition. I'm also puzzled at this focus on conspecific competition, which is not the subject of this manuscript.
-Most of the introduction is focused on indirect effects that competitors may have (altering the environment or genetic variation, etc.), but what about the simpler direct effects of competitors imposing selection?
-Methods, line 102. It would be worth commenting in the Discussion on why evolution was able to occur so quickly in this experiment (~25 generations), given what seems like extraordinarily low genetic variation in T. urdicae. My understanding is that the culture was started with two females, maintained through inbreeding and kept at low densities in culture for 15 years. I would expect these to be effectively be clones at this point and homozygotes at most loci. Does the genetic variation come from a high mutation rate in this species? Is there a reason to think genetic variation was maintained over the course of this 15 years?
-Methods, line 132: I would not refer to these replicates as "islands", because they are not islands. Each "replicate" is made of three plants, with presumably some dispersal among plants, so if anything, it's a metapopulation, rather than an island. But then habitat/plants are constantly being removed (along with the individuals on that plant), so there is constant removal from the "island", that is then replaced by new habitat.
-Line 324 and elsewhere. I assume that "original" means the control treatments that were grown on bean. One could also interpret this as the cucumber treatment before being transferred to the fecundity assay. Deciding what terminology to use for the which treatments would be useful and more clear if the control treatment were described in the Methods.

Reviewer 2 ·

Basic reporting

I feel that the authors' basic reporting is good. The manuscript is well-written and with the improvements made after review, it is even better in this regard.

Experimental design

The authors did not address my comment in regards to experimental design, which was my main criticism of the paper. They did a good job of addressing my minor comments, but not this. Therefore, I do not believe that this manuscript should be published in its current form. They simply disagreed without providing a sufficient justification for their disagreement.

Validity of the findings

No additional comment.

Additional comments

I still do not think that it is valid to claim that this work tests ghost competition, because it was not what the experiment was originally designed to test. The hypothesis should be formed a priori and tested, but that is not what is done with this work. Here the authors have formed their hypothesis a posteriori, and I do not think that is valid.

Reviewer 3 ·

Basic reporting

Some of the new material feels a little bit roughly worded, but overall the paper is clear. References appear to be good, with one minor exception: the authors have added Chesson 2000 as their citation for the role of tradeoffs in species coexistence. While this paper does talk about it, the idea was in the literature since at least the 1970s.

The authors have done some restructuring in response to reviews, which I feel substantially improve the flow of the paper. Figures and tables are professional, and the raw data is shared. The results of this paper are self-contained.

While I think figure 3 may be helpful in data exploration and analysis decisions, I don’t believe it provides information that is important to understanding this paper. I would suggest that it be moved to supplements and the schematic be moved to the main text (I found the schematic very helpful). However, this is up to the authors. I will note that the figure caption references black dots and lines in this figure, which were not visible in my version of the manuscript.

Experimental design

The experiment was well designed, and the new structure and schematic help readers to understand an inherently complicated design. Methods appear to be rigorous, and are described sufficiently to replicate.

Validity of the findings

The authors have carried out a series of analyses to uncover demographic and evolutionary effects of “ghost competitors”. In response to reviews, they have clarified several of their statistical methods, and their approaches are largely statistically sound. With a few exceptions, they have satisfied me with their changes or responses. There are several approaches that are problematic, however.

First, I understand the author’s desire to allow for nonlinearity of time by fitting their 5 census times as a categorical variable instead of continuous one. To the extent that they are working with a maximal model and have sufficient data, this is understandable. However, comparing their model fit procedure for time as a factor vs time as a linear term, AIC strongly supports using time as a linear term. To the extent that they are using model-fitting to support/refute hypotheses of the effect of time, it is unwise to use a poor-fitting metric of time and then discard it when it does not fit well. Given the structure of the maximal model (… time + time*density1 + time*density2…), linear time adds 3 parameters to the maximal model, while time as a factor adds 12, which is likely the cause of the difference in AIC. When I test the authors’ model selection using linear time, the best-fitting model for models B1 and B3 include time. If the authors found that nonlinear relationships with time were important, I would be satisfied, but when the data does not support nonlinearity, it is not satisfying to then disregard simpler interpretations of time.

That said, the authors should not use “presence/absence in best-fitting model” as a metric for whether a term matters, which they do on line 318. There are at least two general issues with this: in the case where several covariates are fairly correlated, if only one is present in the final model, that does not demonstrate that the other is not important (since the first is potentially a proxy for the second). The other issue, which is very relevant here, is that “best-fitting” is different from “only reasonable model”. Models within 2 AIC of one another are generally considered statistically indistinguishable, and in some cases there are models including time that are within 2 AIC of the best fitting model. Looking at the Chisq tests of the maximal model (Table 1), we see that time is marginally significant in predicting fecundity on beans. This is very different from time not mattering. In general, the effect (or lack thereof) of a predictor should be evaluated with hypothesis testing, not model selection.

If time is not a useful predictor of adaptation, do the authors think any evolution happened before the first time point? They might speculate that there was an initial, rapid extinction of certain lineages with or without competitors, and this drove fecundity differences.

I felt that the decision to measure demography with fitted carrying capacity/ stable population size (e.g. the “plateau”) was a good solution to measuring population sizes. However, the plateau is referred to in the paper without enough explanation (first referenced line 263). I think this deserves at least a 1-2 sentence explanation. I think it’s fair to determine the cutoff with visual inspection, but a segmented regression (sometimes called broken stick regression) could be used to allow the data to inform the author’s decision of the cutoff for the plateau. If it were me, I would consider using the segmented package in R to find the best breakpoint for the three relevant categories (T. urticae with and without competition, T. ludeni without competition), and ensuring the author’s cutoff for the plateau was after the various estimated break points. Even simply stating that 200 was chosen as a cutoff by visual inspection, and referencing how this compares to the segmented approach would help validate that specific cutoff.

The authors do a nice job of distinguishing between changes in fecundity (which imply evolutionary effects of competitors) and plateau population size. They attribute plateau population size to ecological processes. While this is a reasonable interpretation, another is that plateau population size is a function of carrying capacity, which could be a function of organismal traits. Under this paradigm we would interpret that authors results as selection for growth rate but not carrying capacity in response to ghosts of competitors. I believe there are some standing hypotheses about tradeoffs between resource acquisition and resource efficiency that the authors could then tie their findings into. I think adding some framing in this direction could be interesting, but the decision to do so is in the authors’ hands. However, I think it would be wise to mention the possibility that plateau population size could be the consequence of adaptation, rather than just assuming it is only the consequence of ecological processes.

Additional comments

Overall I think this is a really interesting study. I have a number of suggestions for word changes or potential ambiguities to avoid, but in general I like this manuscript. Please consider the following as “someone was flagging things that might be worth changing” rather than “these things must be done this way or the manuscript will be rejected”.

• I would consider modifying the start of the paragraph starting line 64. The initial sentence implies an assumption that the inferior competitor will always go extinct, but there are a number of mechanisms that allow competitively inferior species to coexist with superior competitors (unless competitively inferior is defined solely as “unable to coexist”). Since this isn’t the focus of this paper or the important part of this paragraph, I think it would be easier to talk about how even when a species is predicted to go extinct, it can still co-occur for many generations
• Line 130-146: I think it would be helpful to actually start by outlining that the authors made “novel host islands” with sticky tape etc (e.g. line 148), and started by placing 2 plants adjacent to each other (e.g. 131)
• Line 175 “approximately” instead of “about”. While I did not track them all, there are other cases where unnecessarily informal language was used. Consider making a pass of edits looking for these sort of cases.
• 195: missing an “a” between “as” and “proxy”
• 200-202: How do bean discs prevent evolution?
• Line 206: I think “specifically” is more appropriate than “especially”
• Lines 248 etc are pretty clearly a consequence of overfitting with the full model: the authors could list that rather than the more generic “convergence problems”.
• The paragraph line 309 is a little bit complicated. The authors repeatedly talk about populations being more or less fecund; while this may be acceptable shorthand, it feels ambiguous whether this is an ecological or evolutionary measurement. What the authors measured was the low-density fecundity of individuals sampled from these populations (well, really, their grandchildren, if I read the methods correctly). And the actual measurements they made were far more interesting than observed births in a census of the population/population size, which is what “population fecundity” might imply. It might be useful to think about if there is a clear way to convey that succinctly. If not, the present approach feels workable.
• Line 356/357 – I’m not sure that it’s correct or necessary to mention where the extra energy is allocated. Simply stating that T. urticae might have higher resource efficiency is both simpler and more true (larger population size could be a consequence of extended lifespan, for example, or reduced resource requirement per individual).
• Line 368: The “we speculate” sentence and the following sentence can be combined by selecting that the competitive pressure letd to the evolution of increased fecundity
• Something that struck me only late in this paper is that the authors measurement of fecundity does a great job of isolating the effect of mite trait evolution from any kind of direct or indirect effect mites have on their host plant. This is a really cool consequence of their experimental design, and it’s something they could emphasis in the methods (unless I missed it, in which case perhaps it would benefit from clearer emphasis).
• Lines 374-375: appears to have some redundant references to the novel host plant
• Paragraph on line 374 confused me. The presence/absence of differences in fecundity was unclear, and seemed to contradict the findings earlier in the paper. I would suggest spending a bit of time clarifying this paragraph
• Line 395: “performance” can mean many things. Did the authors mean mean fecundity? Or final densities?
• Line 399-401: If ecological specialization in this context is niche differentiation, I would expect performance differences would show up in the context of competition, rather than improved performance on their own. This is likely to be most readers’ intuition, so if this is a different dynamic, clarification is needed.
• Line 425 “in creating” rather than “to create”?
• Figure 1A: data overlap with the legend make it hard to read the legend. T. urticae (no comp) is green, not pink, but that is somewhat confusing to determine at first glance.
• Figure 1A: caption talks about grey zone for 95% CI, but not visible to me.
• Figure 1B: is this pooled across all time points? Worth mentioning in caption.
• Caption for fig 2 talks about a significant interaction between density of T. ludeni and fecundity of T. urticae. I think the authors mean significant relationship? Fecundity is the response variable, so interaction is not the right word.
• Table 1, bottom section: the title says without interspecific competitor, but one of the treatments is with vs without competitors. I found this confusing
• In general, the formulas in table 1 are hard to read. Consider putting them on their own line, and perhaps using further abbreviations, labeled in the table caption, so that each formula fits on a single line. I would also appreciate seeing the df of the maximal model
• Table 3 gives us the most relevant information of all the tables – how did parameters differ? I really appreciated having this, but given the number of models run, it would probably be useful to have reminds of the time periods being compared, and the units of the estimates
• Similarly, table 4 has fecundity listed in some sections and not others
• The lines in the R code that read in the data files are missing underscores (I suspect the underscores of the were added by the data repository)
• The selection of error distribution in the code is somewhat dissatisfying. Using the visual tests of goodness of fit for distributions (e.g. "A fitted distribution should be evaluated using graphical methods (goodness-of-fit graphs automatically provided in our package by plotting the result of the fit (output of fitdist() or fitdistcens() and the complementary graphs that help to compare different fits - see ?graphcomp)." ~ fitdistrplus vignette) show the normal distribution as the clear best option for a number of the models. If the authors have decided a priori that they are uncomfortable using the normal distribution because they have non-negative count data (which is a reasonable feeling), they should exclude that as an option in their model fit code rather than include it and then ignore it.

---

## Round 0.3 · accepted · Accept

This is a much improved manuscript and I appreciate your consideration and implementation of the reviewers' comments. This is a fascinating piece of work and I applaud the authors for turning what may have seemed like a failed experiment into a interesting contribution to ecological and evolutionary research.